# Effects of *Sarcoptes scabiei* Translationally Controlled Tumor Protein (TCTP) on Histamine Release and Degranulation of KU812 Cells

**DOI:** 10.3390/ijms232112865

**Published:** 2022-10-25

**Authors:** Ziyi Xu, Yanting Xu, Xiaobin Gu, Yue Xie, Ran He, Jing Xu, Bo Jing, Xuerong Peng, Guangyou Yang

**Affiliations:** 1Department of Parasitology, College of Veterinary Medicine, Sichuan Agricultural University, Chengdu 611130, China; 2Department of Chemistry, College of Life and Basic Science, Sichuan Agricultural University, Chengdu 611130, China

**Keywords:** histamine, KU812 cells, *Sarcoptes scabiei*, translationally controlled tumor protein

## Abstract

Scabies is a common parasitic dermatological infection worldwide that is often neglected. Scabies mites stimulate host inflammatory symptoms via secreted and excreted proteins, which induce basophil and mast cell degranulation and host histamine release. However, the mechanism of degranulation and histamine release is unclear. Moreover, the *Sarcoptes scabiei* translationally controlled tumor protein (TCTP) is predicted as an excreted protein, which may be involved in host inflammatory response regulation. First, we evaluated *S. scabiei TCTP* gene (*SsTCTP*) transcription in larvae, nymphs, and adults by qRT-PCR, and *SsTCTP* transcription was highest in larvae, followed by nymphs. Second, we found that the *S. scabiei* TCTP recombinant protein (rSsTCTP) promoted mice histamine release in vivo by Evans blue Miles assay. Therefore, to further explore the possible role of *S. scabiei* TCTP in host inflammatory response regulation, we established a degranulation model of KU812 cells. The results of the degranulation model suggested that rSsTCTP could induce enhanced degranulation of KU812 cells and increase the secretion of histamine and the expression of IL-4, IL-6, and IL-13 in vitro. In conclusion, we speculate that scabies mites could stimulate host histamine release and Th2 response by excreting *S. scabiei* TCTP.

## 1. Introduction

Scabies is a persistent zoonotic dermatological disease caused by *Sarcoptes scabiei*. Scabies affects approximately 130 million people worldwide every year, with high morbidity and mortality [1]. Additionally, scabies mites can infect 148 species of domestic and wild animals [2], posing a serious threat to human and animal health and causing global public health problems and economic problems. Scabies was named one of the 15 most annoying dermatological diseases in the world and was classified as the highest category of neglected tropical diseases in 2017 [3,4]. 

During the incubation period (4–6 weeks) after the initial scabies infection, the host skin usually shows only redness and swelling; however, as the infection worsens, the symptoms rapidly develop into itching and skin lesions, most commonly small scattered papules [5,6]. This is related to the parasitic pattern of scabies mites; the host immune response is inhibited in the early infection period via proteins secreted and excreted by *S. scabiei* [7,8]. However, after 4–6 weeks, the inflammatory response to scabies intensifies, with abundant leakage of lymph and plasma, which are conducive to the intake of nutrients by scabies mites [3,9]. This inflammatory response is inseparable from the effect of host basophil and mast cell degranulation, and in scabies skin lesions, they have similar morphologies and functions [10,11]. Mast cells and basophils are activated to rapidly release histamine, which plays a role in the acute symptoms of inflammation, including vasodilation, enhanced smooth muscle contraction, nerve reflex, and vascular leakage, leading to tissue edema [12]. The host is also stimulated to produce Th2 cytokines in the late stage of scabies, which are responsible for IgE-mediated allergic inflammation [13,14]. However, it remains unclear how scabies mites regulate host mast cell and basophil degranulation and histamine release. Studies on histamine and Th2 cytokines in primary basophils are difficult and time-consuming because the percentage of this cell population in peripheral blood is extremely low. In addition, for mast cells and basophils, differences in the immune systems of donors often lead to inconclusive or conflicting outcomes, whereas cell lines passaged in vitro are more stable [15]. In a recent study, we found a human leukemia cell line, KU812, that has similar biological characteristics and functions to mast cells and basophils and may serve as an appropriate model to study degranulation [16]. Its fast and specific response to biological factors makes it suitable as a biological assay for the determination of active factors produced by atopic individuals, and its degranulation is characterized by a rapid increase in histamine, which can better reflect the degree of inflammatory reaction [17,18].

Translationally controlled tumor protein (TCTP) is also known as histamine-releasing factor (HRF). The homologous protein in the host is a housekeeping protein, which mediates microtubule stability, calcium-binding activity, and apoptosis, and it interferes with the extension of intracellular proteins [19]. In the extracellular matrix, TCTP is mainly in charge of the IgE-mediated inflammatory response, inducing Th2 cytokines and chemokines, and enhancing B cell proliferation in the late stage of inflammation [20]. Moreover, homologous TCTPs from parasites have been reported to promote host histamine release and degranulation of mast cells and basophils. High concentration levels of histamine and *Plasmodium falciparum* TCTP have been found in the serum of patients, and recombinant *P. falciparum* TCTP can stimulate histamine release from basophils in vitro [21,22,23]. TCTPs from *Brugia Malayi*, *Wuchereria Bancrofti*, *Schistosoma Mansoni*, *Schistosoma japonicum*, *Ixodes Scapularis*, and *Dermanyssus gallinae* have been shown to stimulate mast cells to release histamine in vitro [24,25,26,27,28,29]. *Dermacentor variabilis* TCTP can promote histamine release and increases blood flow in the feeding sites of ticks, thus facilitating nutritional intake and growth [27,28]. In the latest genome database of *Sarcoptes scabiei*, TCTP is predicted to be an excreted protein [30]; however, its role is unknown. In addition, we found that the *S. scabiei* TCTP recombinant protein (rSsTCTP) promoted mouse histamine release in vivo by Evans blue Miles assay. Therefore, to further explore whether and how *Sarcoptes scabiei* TCTP (SsTCTP) is involved in regulating the host inflammatory reaction, we used KU812 cells as a degranulation model to assess the effect on mast cells and basophils. Finally, the histamine release and cytokines-like activity of SsTCTP were determined, deepening our understanding of how scabies mites regulate host mast cell and basophil degranulation and histamine release.

## 2. Results

### 2.1. Cloning, Expression, and Purification of SsTCTP

A cDNA fragment encoding the predicted amino acid sequence was amplified by PCR using a pair of primers designed using existing expressed sequence tag (EST) libraries. Sequencing confirmed that the *SsTCTP* clone was 100% identical to the published GenBank sequence (accession number: QR98_0093560). The protein was expressed in *Escherichia coli* BL21 cells, isolated from inclusion bodies under denaturing conditions, and purified using a Ni column. Highly pure SsTCTP was demonstrated using Coomassie blue-stained gel analysis (Figure 1).

### 2.2. qRT-PCR Analysis of Stage-Specific SsTCTP Relative Transcription

Initial primer testing and optimization of PCR for the amplification of stage-specific *SsTCTP* sequences was carried out. No cross-amplification indicated that the primers were sequence-specific. Comprehensive analysis using *SsGAPDH* as the reference gene showed that the transcription levels of *SsTCTP* were different at each stage, and the relative transcription level in larvae and nymphs was significantly higher than that in adults (*p* < 0.0001) (Figure 2).

### 2.3. Effect of rSsTCTP on Stimulating Mice Vascular Hyperpermeability

Compound 48/80 (C48/80, a polymer produced by condensation of 4-methoxyphenylethylamine and formaldehyde), is a histamine liberator and mast cell degranulation agent [31] that can be used as a positive control to stimulate host vascular hyperpermeability in the Evans blue Miles assay. With the endogenous histamine inhibitor pyrilamine maleate inhibiting the release of histamine in vivo, host histamine release only depended on the samples. The results revealed that the group with rSsTCTP stimulation had significantly increased vascular permeability of subcutaneous tissues compared with the negative control group (*p* < 0.05), and the effect was not statistically different from that in the positive control group (*p* > 0.05) (Figure 3). These results suggest that rSsTCTP can stimulate vascular hyperpermeability in mice by increasing the release of histamine in vivo.

### 2.4. Effect of rSsTCTP on KU812 Cell Proliferation

The effect of rSsTCTP on KU812 cell proliferation was detected using the CCK-8 assay to determine the optimal reaction concentration for subsequent experiments. When the co-incubation time was 1 h, compared with the no-treatment control, the proliferation of KU812 cells was significantly inhibited by 8 µg/mL rSsTCTP (*p* < 0.05) and 80 µg/mL C48/80 (*p* < 0.01). PBS, pET-32α, 50 μg/mL C48/80, and 2 μg/mL and 4 μg/mL rSsTCTP induced relatively little inhibition of KU812 cell proliferation (*p* > 0.05). The proliferation rate of the no-treatment control was set to 1, and the average proliferation rates were 0.943 and 0.963 when stimulated by 50 µg/mL C48/80 and 2 µg/mL rSsTCTP, respectively (Figure 4). Therefore, to exclude the influence of cell proliferation on subsequent experiments, 2 µg/mL rSsTCTP and 50 µg/mL C48/80 were selected as the optimal reaction concentrations.

### 2.5. rSsTCTP Activates the Lyn-Mediated Signaling Pathway in KU812 Cell Degranulation

Western blotting analysis of the Lyn-mediated signaling pathway in KU812 cell degranulation detected the levels of phosphorylated Lyn, NF-κB, p38 MAPK, JNK1/2, and ERK1/2 proteins (Figure 5A). Compared with those in the PBS group, the levels of phosphorylated Lyn, NF-κB, p38 MAPK, JNK1/2, and ERK1/2 were increased to varying degrees by rSsTCTP stimulation (*p* < 0.01) (Figure 5B). Between the pET-32α group and the PBS group, there was no significant difference (*p* > 0.05). Therefore, we speculated that rSsTCTP could stimulate KU812 cells to degranulate by activating the downstream NF-κB and MAPK family phosphorylation of the Lyn-mediated signaling pathway, finally releasing degranulated products. 

### 2.6. Stimulatory Effects of rSsTCTP on β-HEX and Histamine Release from KU812 Cells

β-Hexosaminidase (β-HEX) and histamine are biomarkers of the degranulation level [32,33]. Compared with the PBS and pET-32α groups, the rSsTCTP-induced stimulation and positive control Compound 48/80 significantly increased β-HEX release (*p* < 0.0001), and the discharge rates were above 70% (Figure 6A). The results showed that rSsTCTP and Compound 48/80 could significantly stimulate KU812 cells to degranulate. Combined with the observation of increased phosphorylation levels of relevant proteins, the degranulation model of KU812 cells was successfully constructed.

rSsTCTP significantly increased the release of histamine compared with that of the PBS and pET-32α groups (*p* < 0.0001, *p* < 0.001) (Figure 6B). These results were consistent with those of the Evans blue Miles assay in vivo. These results in KU812 cells suggested that rSsTCTP might induce mast cells and basophils to degranulate and then release histamine, thus increasing the permeability of the subcutaneous tissue and the leakage of plasma and lymph.

### 2.7. rSsTCTP Induces the Enhancement of the Th2-Type Immune Response

The Lyn-mediated signaling pathway in KU812 cell degranulation is followed by cross-linking activation of FcεRI, which binds to IgE and causes a Th2-type immune response in scabies [13,14]. Compared with those in the PBS group, the release levels of Th2 cytokines IL-4, IL-6, and IL-13 increased to varying degrees under rSsTCTP stimulation (*p* < 0.0001) (Figure 7), indicating that rSsTCTP can upregulate the host Th2 immune response through the Lyn-mediated signaling pathway in degranulation.

## 3. Discussion

Firstly, we cloned and expressed the TCTP gene of *Sarcoptes scabiei* and analyzed its transcription level in larvae, nymphs, and adults. The relative transcription levels of SsTCTP were different at each stage, with the transcription level in larvae and nymphs being significantly higher than that in adults. However, in ticks, TCTP genes are expressed in immature and mature stages, and their relative transcription levels did not correlate with developmental stages [26,27]. The difference in TCTP transcription levels between the two ectoparasites (ticks and mites) might be related to the parasites’ characteristics. Scabies mites parasitize subcutaneous tissue and directly contact interstitial fluid. At the same time, the difference between them may also be related to the tertiary structure of TCTP-binding proteins and their counterparts. Previous studies have shown that variations in TCTP may correlate with its binding partners and, consequently, the functional roles of TCTP-binding proteins [34]. In addition, the varying transcription levels of SsTCTP in different stages may be due to the different proteins required to bind, which is also an adaptation for survival.

In vivo, Evans blue is a dye with a high affinity for plasma albumin. When vascular permeability is increased, Evans blue penetrates tissues along with the plasma and can be observed [35]. Histamine is a strong vasodilator that can increase the permeability of the capillaries and cause plasma leakage into the tissues, resulting in local tissue edema. With the endogenous histamine inhibitor pyrilamine maleate inhibiting the release of histamine [35], the increased leakage of Evans blue indicated that rSsTCTP directly increased the release of histamine in vivo. Similar to ticks [26,27,28], scabies mites feed on the lymph and plasma [36], and SsTCTP might increase the leakage of lymph and plasma by stimulating the host’s release of histamine, which is conducive to nutritional intake and parasitism of scabies mites. Therefore, to further explore the possible role of *S. scabiei* TCTP in host inflammatory response regulation, we established a degranulation model of KU812 cells.

After KU812 cells were stimulated rSsTCTP, a phosphorylated tyrosine kinase (Lyn) activated downstream signaling proteins in the degranulation pathway, resulting in the phosphorylation of NF-κB and the MAPK family and the synthesis and release of cell particles and inflammatory factors. Among them, histamine and β-HEX are biomarkers of the degranulation reaction of mast cells and basophils [32,33]. The increased release of histamine and β-HEX and increased levels of phosphorylated Lyn, NF-κB, and the MAPK family indicated that the degranulation model of KU812 cells was successfully constructed, and rSsTCTP could induce enhanced degranulation of KU812 cells. Moreover, the increase in histamine release from KU812 cells was also confirmed by the in vivo results; thus, rSsTCTP possesses histamine-releasing factor activity.

Lyn-mediated signaling in KU812 cell degranulation is followed by cross-linking activation of FcεRI, which binds to IgE and causes a Th2 immune response [12,13]. The ELISA results showed that the IL-4, IL-6, and IL-13 levels increased to varying degrees, indicating that rSsTCTP may be involved in the upregulation of the Th2 immune response in scabies. Th2-type cutaneous inflammation is associated with repeated, intense itching and skin lesions [37,38]. SsTCTP might aggravate the Th2 inflammatory response in scabies, further aggravating the clinical symptoms.

In conclusion, rSsTCTP can stimulate the release of histamine and promote membrane permeability to lymph and plasma, which might be conducive to the nutrition intake of scabies mites. At the same time, rSsTCTP could increase IL-4, IL-6, and IL-13 to varying degrees, indicating that rSsTCTP might enhance the Th2 inflammatory response, which is related to the allergic inflammatory lesions of scabies. We speculate that scabies mites can upregulate the degranulation of mast cells and basophils through SsTCTP and ultimately affect the host inflammatory response.

## 4. Materials and Methods

### 4.1. Animal Ethical Statement

This animal study was reviewed and approved by the Animal Care and Use Committee of Sichuan Agricultural University (SYXK 2019–189). All procedures involving relevant experimental animals were carried out strictly in accordance with the Guide for The Care and Use of Laboratory Animals (National Research Council, Bethesda, MD, USA) and the Animal Research: Reporting of In Vivo Experiments (ARRIVE) guidelines (http://www.nc3rs.org.uk/arrive-guidelines, accessed on 20 April 2021). The care and use of animals were carried out in accordance with applicable institutional and national guidelines.

### 4.2. Mites, Experimental Animals, and Cells

Four female rabbits (*Oryctolagus cuniculus*) weighing about 2.5 kg were artificially infected with *S. scabiei* from the Department of Parasitology, Sichuan Agricultural University. Mites were collected from the skin lesions of the feet of rabbits under the same feeding conditions.

Six 8-week-old specific pathogen-free (SPF) female Kunming mice were purchased from Dashuo Laboratory Animal Co., Ltd. (Chengdu, China). The animals were kept at room temperature and were given ad libitum access to water and food.

The human peripheral blood basophilic leukemia cell line (KU812) was purchased from ATCC (Manassas, VA, USA). KU812 cells were incubated in a 37 °C-preheated Roswell Park Memorial Institute (RPMI) 1640 cell culture medium with 10% fetal bovine serum, 100 U/mL penicillin, and 100 μg/mL streptomycin at 5% CO_2_ and 37 °C.

### 4.3. Preparation of Recombinant S. scabiei TCTP (rSsTCTP)

The coding sequence (CDS) of the *SsTCTP* (QR98_0093560) was amplified from the *S. scabiei* cDNA library by PCR using a pair of primers containing the *BamHI* and *HindⅢ* restriction sites: 5′-CGGGATCC-ATGATCATCTACAAGGATTT-3′ and 5′-CCAAGCTT-TCACACTTTTTCAGCTTC-3′; it was ligated into plasmid pET-32α(+) to construct recombinant plasmid pET-32α(+)-SsTCTP. The plasmid was transformed into *Escherichia coli* BL21 cells (Takara, Shiga, Japan), and expression was induced using isopropyl-β-D-thiogalactopyranoside (IPTG). The recombinant protein was isolated from inclusion bodies under denaturing conditions and purified using a nickel (Ni) column. Its concentration was determined using bicinchoninic acid (BCA) assay, and the ToxOut™ High-Capacity Endo-toxin Removal Kit (Thermo Fisher, Shanghai, China) was used to remove endotoxins to less than 0.1 EU/mL, as detected by the ToxinSensor™ Chromogenic LAL Endotoxin Assay Kit (Thermo Fisher, Shanghai, China). The recombinant protein was stored at −80 °C.

### 4.4. Analysis of SsTCTP Transcription at Each Developmental Stage

The mixed *Sarcoptes scabiei* samples were classified according to Arlian [39], and RNA was extracted from larvae, nymphs, and adults (Tiangen Biotech, Beijing, China). The RNA was reverse-transcribed to cDNA, which was used as the template in the quantitative PCR (qPCR) step of the quantitative real-time reverse transcription PCR (qRT-PCR) protocol to quantify *SsTCTP* expression using *Ss18sRNA* and *SsGAPDH* (encoding glyceraldehyde-3-phosphate dehydrogenase) as controls. The reaction system and conditions were set according to the Tiangen qRT-PCR kit instructions (Beijing, China). ∆∆Ct (Qr = 2^−∆∆Ct^) was used to analyze the relative transcription [40] of *SsTCTP* in different development stages. The primers used are shown in Table 1.

### 4.5. Evans Blue Miles Assay

To investigate the effect of rSsTCTP on histamine release in vivo, an Evans blue vascular permeability mouse model was adopted. First, 24 h before stimulating vascular hyperpermeability, six mice were anesthetized separately using 3% isoflurane with 1.5% isoflurane for anesthesia maintenance. Under anesthesia, avoiding injuries to the skin, both flanks of each mouse were carefully shaved. Separate sterile solutions of 4 µg/µL of the histamine inhibitor, pyrilamine maleate, and of 1% *w/v* Evans blue dye (Gibco, Shanghai, China) were prepared and sterilized through a 22 µm filter in a laminar flow cabinet. Then, each mouse was intraperitoneally injected with pyrilamine maleate at 10 µL/gram of body weight using a sterile 1 mL syringe with a 30 G needle. Using a suitable heat lamp to promote vasodilation, the tail of each mouse was intravenously administered with 100 µL of Evans blue dye. After the dye circulated for 30 min, the first mouse tested was anesthetized with isoflurane and then intradermally injected with 20 µL of rSsTCTP, phosphate-buffered saline (PBS), pET-32α, and Compound 48/80 (1 μg, C48/80, promotes histamine release) into the flank; each injection site was at least 1 cm apart. The mouse was sacrificed 20 min after the injections by cervical dislocation; a vertical incision of 3–4 cm was made in the flank skin, and then the skin was teased away from both flanks to reveal the sites of Evans blue accumulation. 250 µL of deionized formamide was used to extract the Evans blue dye, 100 µL of which was added into a separate well of a 96-well plate, and the peak Evans blue absorbance at 620 nm was measured with a reference reading of 740 nm.

### 4.6. Cell Proliferation Assay

First, 100 µL of KU812 cell suspension (1 × 10^6^ cells/mL) was stimulated with PBS or pET-32α (10 µg/mL), respectively, in the presence of different concentrations of rSsTCTP (2, 4 and 8 µg/mL) and Compound 48/80 (50 and 80 µg/mL). Then, 10 µL of Cell Counting Kit-8 (CCK-8) solution (Thermo Fisher, Shanghai, China) was reacted with the cells in each well and incubated for 2 h, after which the absorbance at 450 nm was determined.

### 4.7. Western Blotting Analysis

Protease inhibitor (4 µL) and phosphatase inhibitor (4 µL) were added to 1 mL of total radioimmunoprecipitation assay (RIPA) protein extraction solution (Gibco, Shanghai, China) and fully mixed. Then, 200 µL of the prepared protein extracting solution was added to 10^6^–10^7^ KU812 cells to fully lyse them. After centrifugation at 200× *g* at 4 °C for 15 min, the supernatant was transferred to another tube. SDS-PAGE was used to separate 50 µg of total protein, followed by electrotransfer onto nitrocellulose filter membranes (NC membranes). The NC membranes were incubated in 5% skim milk at room temperature for 2 h. Separate NC membranes were incubated with different polyclonal antibodies (anti-β-actin, 1:8000; anti-nuclear factor kappa B (NF-κB), 1:4000; anti-phosphorylated (p)-NF-κB, 1:2000; anti-LYN proto-oncogene, Src Family tyrosine kinase (Lyn), 1:2000; anti-p-Lyn, 1:2000; anti-Jun N-terminal kinase 1/2 (JNK1/2), 1:1000; anti-p-JNK1/2, 1:2000; anti-mitogen-activated protein kinase 14 (MAPK14) (p38 MAPK), 1:2000; anti-p-p38 MAPK, 1:2000, anti-extracellular regulated kinase 1/2 (ERK1/2), 1:2000, and anti-p-ERK1/2, 1:2000) (all ABclonal, Wuhan, China) diluted in 5% skim milk overnight at 4 ℃. After washing three times, the membranes were incubated in horseradish peroxidase (HRP)-conjugated secondary antibody (1:8000, diluted with 5% skim milk) (ABclonal, Wuhan, China) at room temperature for 2 h. Then, after washing three times, the membranes were developed using an ultra-sensitive ECL chemiluminescence kit (Tiangen Biotech, Beijing, China). Finally, the immunoreactive protein bands were quantified using Image J (NIH, Bethesda, MD, USA).

### 4.8. β-Hexosaminidase (β-HEX) Release Assay

KU812 cells were seeded in 6-well plates (5 × 10^5^ cells/well) at 37 ℃ in 5% CO_2_ for 20 h. The cells were treated with PBS, pET-32α, C48/80, or rSsTCTP for 1 h. The supernatants (50 µL) were removed and incubated with a substrate buffer (3.3 mM β-D-glucopyranoside, pH 4.5) in a 96-well plate at 37 °C for 1 h. The reaction was terminated using 100 µL of stop solution (0.1 M sodium carbonate (Na_2_CO_3_)/sodium bicarbonate (NaHCO_3_), pH 10.0), and the absorbance was measured at 405 nm. The β-HEX release activity was determined as follows: β-HEX release activity (%) = (OD_405_ of sample − OD_405_ of blank/OD_405_ of control − OD_405_ of blank) × 100.

### 4.9. Enzyme-Linked Immunosorbent Assay (ELISA) Detection of Histamine and Th2 Cytokines

Histamine, IL-4, IL-6, and IL-13 were detected according to the ELISA kit instructions (Multi Sciences, Hangzhou, China). Briefly, KU812 cells were seeded in 6-well plates (5 × 10^5^ cells/well) at 37 °C in 5% CO_2_ for 20 h and then stimulated with PBS, pET-32α, C48/80, or rSsTCTP for 1 h. The histamine, IL-4, IL-6, and IL-13 concentrations were measured according to their absorbance at 450 nm.

### 4.10. Statistical Analysis

All data are expressed as the means ± the standard deviation (SD). GraphPad Prism 5 (GraphPad Inc., La Jolla, CA, USA) was used to perform the statistical analysis. Differences between groups were assessed using one-way analysis of variance (ANOVA) in SPSS 11.5 (IBM Corp., Armonk, NY, USA). Differences were considered statistically significant at *p* < 0.05.

## Figures and Tables

**Figure 1 ijms-23-12865-f001:**
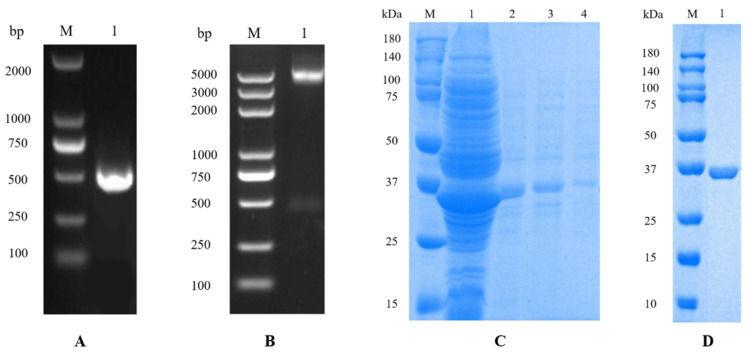
Gene cloning, expression, and protein purification of SsTCTP. (**A**): PCR amplification of the *SsTCTP* gene (M: gene marker DL2000; 1: PCR product); (**B**): identification of recombinant pET-32α-SsTCTP using double digestion (M: gene marker DL5000; 1: recombinant plasmid digested by *BamHI* and *HindШ*); (**C**): solubility analysis of rSsTCTP (M: protein marker; 1: supernatant; 2: 2 M urea dissolved; 3: 4 M urea dissolved; 4: 8 M urea dissolved); (**D**): purification of rSsTCTP (M: protein marker; 1: purification of rSsTCTP). SsTCTP, *Sarcoptes scabiei* translationally controlled tumor protein; PCR, polymerase chain reaction; rSsTCTP, recombinant SsTCTP.

**Figure 2 ijms-23-12865-f002:**
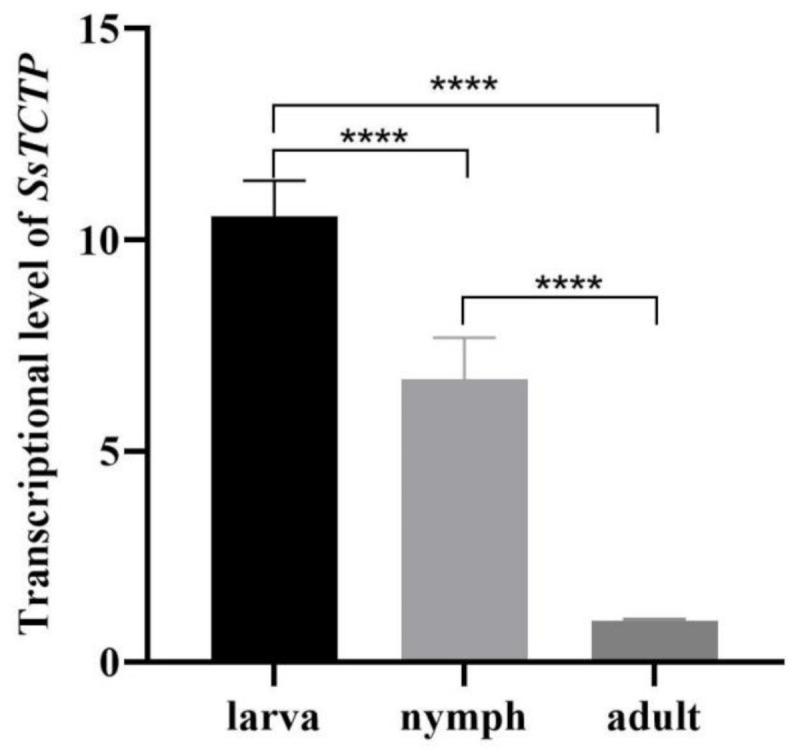
Transcription of the *SsTCTP* at larvae, nymphs, and adults. Data from three individual experiments (each experiment had three replicates) are shown as the means ± the standard deviation (SD). Statistical differences between groups were calculated using Student’s *t*-test (**** *p* < 0.0001). *SsTCTP*, *Sarcoptes scabiei* translationally controlled tumor protein gene.

**Figure 3 ijms-23-12865-f003:**
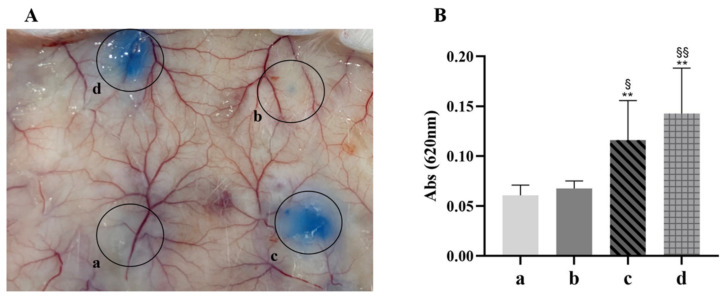
Evans blue aggregation of mouse dermis (**A**) and Evans blue absorption peak at 620 nm (**B**). a: PBS, b: pET-32α, c: rSsTCTP, d: Compound 48/80. (**A**) A representative result of six independent mouse experiments. (**B**) Evans blue extravasation was estimated after extraction with formamide and reading at an absorbance 620 nm. Data are shown as the means ± the standard deviation (SD) of six mice. ** *p* < 0.01 versus PBS group; ^§^
*p* < 0.05, ^§§^
*p* < 0.01 versus pET-32α group (*t*-test). rSsTCTP, recombinant *Sarcoptes scabiei* translationally controlled tumor protein; PBS, phosphate-buffered saline.

**Figure 4 ijms-23-12865-f004:**
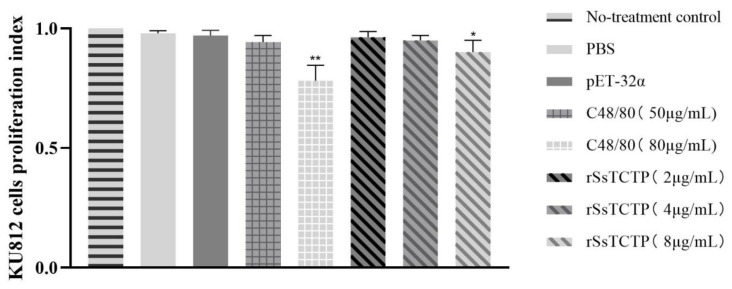
Effects of PBS, pET-32α, C48/80, and rSsTCTP on KU812 cell proliferation. Data from three individual experiments (each experiment had three replicates) are shown as the means ± the standard deviation (SD). * *p* < 0.05, ** *p* < 0.01 versus no-treatment control (*t*-test). rSsTCTP, recombinant *Sarcoptes scabiei* translationally controlled tumor protein; PBS, phosphate-buffered saline.

**Figure 5 ijms-23-12865-f005:**
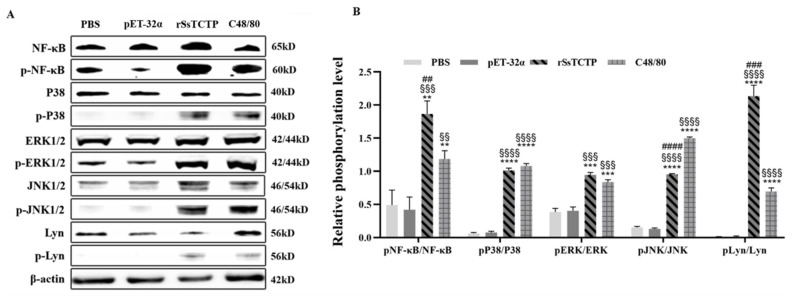
Protein levels of phosphorylated (p) NF-κB/NF-κB, pP38/P38, pERK/ERK, pJNK/JNK, and pLyn/Lyn in KU812 cell degranulation. (**A**) Representative Western blotting of three individual experiments. (**B**) The graph of the quantified band density. Data are shown as the means ± the standard deviation (SD) of three Western blotting analyses. ** *p* < 0.01, *** *p* < 0.001, **** *p* < 0.0001 versus PBS group; ^§§^
*p* < 0.01, ^§§§^
*p* < 0.001, ^§§§§^
*p* < 0.0001 versus pET-32α group; ^##^
*p* < 0.01, ^###^
*p* < 0.001, ^####^
*p* < 0.0001 versus C48/80 group (*t*-test). NF-κB, nuclear factor kappa B; p38, mitogen-activated protein kinase 14 (MAPK14); ERK, extracellular regulated kinase; JNK, JUN N-terminal kinase; Lyn, LYN proto-oncogene, Src Family tyrosine kinase. rSsTCTP, recombinant *Sarcoptes scabiei* translationally controlled tumor protein.

**Figure 6 ijms-23-12865-f006:**
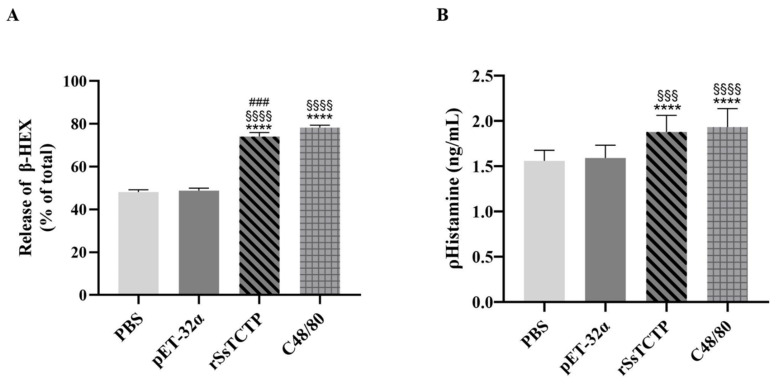
Release of β-Hexosaminidase (β-HEX) (**A**) and histamine (**B**) from KU812 cells. Data from three individual experiments (each experiment had five replicates) are shown as the means ± the standard deviation (SD). **** *p* < 0.0001 versus PBS group; ^§§§^
*p* < 0.001, ^§§§§^
*p* < 0.0001 versus pET-32α group; ^###^
*p* < 0.001 versus C48/80 group (*t*-test). rSsTCTP, recombinant *Sarcoptes scabiei* translationally controlled tumor protein.

**Figure 7 ijms-23-12865-f007:**
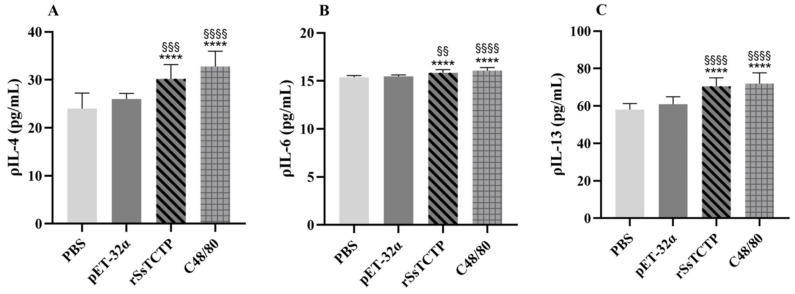
Release of interleukin (IL)-4 (**A**), IL-6 (**B**), and IL-13 (**C**) from KU812 cells. Data from three individual experiments (each experiment had five replicates) are shown as the means ± the standard deviation (SD). **** *p* < 0.0001 versus PBS group; ^§§^
*p* < 0.01, ^§§§^
*p* < 0.001, ^§§§§^
*p* < 0.0001 versus pET-32α group (*t*-test). rSsTCTP, recombinant *Sarcoptes scabiei* translationally controlled tumor protein.

**Table 1 ijms-23-12865-t001:** Primers used in qRT-PCR.

Target Gene ^†^	Primer Sequences (5′-3′)
Ss-TCTP	F: CGGTTGGTTGGTGATTGCATGTG
	R: CGACCTCTTCAGCGGATGGATTG
Ss-18sRNA	F: ATTTGGGTTGCGGATCAGGTCTAAG
	R: CACCACTTTCTGTTTCCCGTTCAAG
Ss-GAPDH	F: CCGTCACAGCCACTCAGAAACC
	R: ACCAGTTGAAGCGGGGATGATATTC

^†^ SsTCTP, *Sarcoptes scabiei* translationally controlled tumor protein; GAPDH, glyceraldehyde-3-phosphate dehydrogenase.

## Data Availability

The datasets supporting the conclusions of this study are included in this article.

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
