# Peer review of "Effects of Sarcoptes scabiei Translationally Controlled Tumor Protein (TCTP) on Histamine Release and Degranulation of KU812 Cells"

_ijms, 2022, doi:10.3390/ijms232112865_

Round 1
Reviewer 1 Report
This is an interesting and well conducted study into the function of TCTP excreted by Scabies. The authors used both in vivo and in vitro techniques to investigate the mechanism of action of the protein. Using in vivo experiments in mice, they compared the vascular hyperpermeability response to recombinant scabies TCTP to that induced by the known compound C48/80. Using the cell line KU812 they determined the effect of TCTP on signaling pathways leading to histamine release.
The results support a pathogenic role of scabies TCTP on the stimulation of histamine release from basophils and mast cells.
Minor comment: The authors state that the scabies TCTP is 100% identical to the GenBank sequency NP_650048. This accession number belongs to TCTP from drosophila melanogaster. It is of interest that while this sequence shares only ~50% sequence homology to human or mouse TCTP, it is still able to stimulate both mouse and human pathology. It may be of interest to the reader to refer to the recent publication by Koo N et al in PLoS One (Comprehensive analysis of Translationally Controlled Tumor Protein (TCTP) provides insights for lineage-specific evolution and functional divergence), or similar publications.
Reviewer 2 Report
The study by Xu et al addresses an interesting topic on the effect of TCTP on the histamine release in vivo and in vitro. The experiments appear to be of good quality and the manuscript is, in general, well written and reads easily. There are, however, some points which need to be addressed because it would enhance the overall quality of the paper.
1.- Figure 3 is counter-intuitive. Readers probably would expect a bar graphics where the values of the untreated controls would be on the far left and the experimental samples on the right as they did, for instance in Figure 5. Because the figure follows the order of the in vivo image (3A), the resulting graph (3B) has problem-control-control-problem. This order is truly hard to follow. The in vivo image should be re-labelled to put a and b as controls and c and d as samples (even if they are apart from each other in 3A). This would make the interpretation of the results easier.
2.- Throughout the figure legends, the authors use the word “replicate”. This term is confusing because it is mostly used to define a reproduction of a group but within a single experiment (i.e., duplicates, triplicates). Authors should clarify if they mean that data came from an individual experiment, performed in triplicate (i.e. at the same time), or rather they refer to data obtained from three independent experiments (performed at different days). This distinction is important.
3.- The title refers to “host histamine release”. The truth is that only one experiment was performed on animals (figure 3) and the results appear somehow isolated and out of place. The term “host histamine release” should be therefore removed from the title, as it is an overstatement not clearly supported by their data (one single experiment is not enough). I suggest that the rationale for the in vivo experiment, within this context, should be better clarified or even entirely removed. If, however, the authors decide to keep that experiment, pictures from at least two more animals should be provided. The SD on the experimental groups of 48/80 and RsTCTP are very high (3B) and those deviations should correlate with the in vivo experiments (3A). Finally, the authors should revise if the results of compound 48/80 are truly not statistically significant over the controls, since they appear to be similar to those of rSsTCTP (which are).
4.-The main problem of the paper, however, is the discussion. It is very long and out focus. In fact, many sentences included there are not suitable for the discussion, but rather they would belong to the introduction as only refer to generalities that do not help to put their findings within the general context of science (lines 171-224). Therefore, the discussion must be completely revised and re-written to properly put their findings (and only their findings) within the general context of the field.
Other minor points are:
5.- Since KU812 cells are the main experimental model, the authors should provide some more details explaining why they chose these cells over others.
6- Figures 4, 5B, 6 and 7. Representation of statistical significance (stars) should clearly indicate what is the group of reference. I presume that it would be the PBS group, but this should be clearly indicated in the figure legend.
6.- The English is very good. I found, however, one typo in the Abstract (line 15). It should read may BE involved.
Round 2
Reviewer 2 Report
The authors have modified the article according to my queries and at the surface the paper seems improved. One of the clarifications I requested, however, has brought into light a critical flaw in the design of the study. All data contained in figures 2, 4, 5, 6 and 7 come from a SINGLE experiment. This is completely unnacceptable. Setting replicates within an experiment is a must, but also it is to have at least three independent experiments before reaching any meaningful conclusion. Working with cell lines is difficult, and the culture conditions of that particular day may have a great influence on the results. This is why independent experiments (not only replicates) have to be performed, to ensure reproductibility of the results, and the statistical analysis should be carried out over the data obtained in all independent experiments.
Even though the data are of potential interest, the study needs at least two more experiments for the indicated figures because I think that we cannot draw any conclusions based on just one experiment.
Round 3
Reviewer 2 Report
c
Author Response
We sincerely thank you for your positive comments on this paper.